# Bowel Health in U.S. Vegetarians: A 4-Year Data Report from the National Health and Nutrition Examination Survey (NHANES)

**DOI:** 10.3390/nu14030681

**Published:** 2022-02-06

**Authors:** Maximilian Andreas Storz, Gianluca Rizzo, Alexander Müller, Mauro Lombardo

**Affiliations:** 1Department of Internal Medicine II, Center for Complementary Medicine, Freiburg University Hospital, Faculty of Medicine, University of Freiburg, 79106 Freiburg, Germany; alexander.mueller@uniklinik-freiburg.de; 2Independent Researcher, Via Venezuela 66, 98121 Messina, Italy; gianlucarizzo@email.it; 3Department of Human Sciences and Promotion of the Quality of Life, San Raffaele Roma Open University, 00166 Rome, Italy; mauro.lombardo@uniroma5.it

**Keywords:** vegetarian, plant-based, fiber, bowel health, constipation, NHANES, bristol stool scale, stool

## Abstract

Dietary fiber is of paramount importance in the prevention of large-bowel diseases, yet fiber intake in many high income countries is well below daily recommendations. Vegetarian diets high in fiber-rich plant-foods have been associated with a higher frequency of bowel movements and softer stools. Thus, vegetarians appear to suffer less frequently from constipation and other bowel disorders. The number of studies investigating these associations, however, is limited. The present study sought to investigate bowel health and constipation prevalence in a self-identified vegetarian population from the U.S. National Health and Nutrition Examination Survey (2007–2010). Bowel health assessment included Bristol Stool Scale (BSS), Bowel Movement (BM) frequency and Fecal Incontinence Severity Index (FISI). The present study included 9531 non-vegetarians and 212 vegetarians. We found no associations between vegetarian status and all examined bowel health items (BM frequency, BSS and FISI). Vegetarians consumed significantly more fiber than omnivores (21.33 vs. 16.43 g/d, *p* < 0.001) but had a lower moisture intake (2811.15 vs. 3042.78 g/d, *p* = 0.045). The lack of an association of vegetarian status and bowel health is surprising, and may be a result of the relatively low fiber intake in this particular vegetarian cohort, which did not meet the daily fiber recommendations.

## 1. Introduction

Four decades ago, the famous Irish surgeon Denis P. Burkitt and his team recognized the important role of dietary fiber in the prevention of certain large-bowel diseases [1,2], which by now have become highly prevalent in the Western hemisphere [3,4,5]. Diets high in fiber result in large, soft stools that traverse the intestine rapidly, whereas fiber-deficient diets abundant in refined foods were linked to constipation and colon cancer [6].

Currently, dietary fiber intake in high income countries is approximately 15 g/day, and, as such, well below the amount of fiber Burkitt advocated for (>50 g/day) [5]. Diets low in fiber avoid or minimize beans, nuts, peas, lentils, legumes, brown rice, whole grains, nuts and seeds and emphasize well-cooked red meat, fish, poultry, eggs and certain dairy products [7]. Such diets have been occasionally recommended following bowel surgery or treatments (such as radiotherapy) that damage the digestive tract [8]. In many individuals, dietary fiber restriction results in a reduction in the size and frequency of stools [9].

Plant-based diets (including vegetarian and vegan diets), in contrast, are rich in fiber-dense foods, including fruits, legumes and green leafy vegetables [10]. These diets influence gut motility and the composition of the human gut microbiota [11,12,13,14]. Although studied less frequently, several trials also suggested differences in defecation and stool patterns between vegetarians and non-vegetarians [11].

Vegetarian diet has been associated with a higher frequency of bowel movements and softer stools with fewer superficial cracks [15]. As such, vegetarians appear to be less frequently affected by (atonic) constipation [11,16], a widespread clinical condition that may negatively impact quality of life to a similar extent as musculoskeletal disorders, allergies and chronic inflammatory bowel disorders [17]. Some studies also suggested an inverse dose relationship between the amount of meat in diet and the frequency of defecation [18]. Vegetarian diets avoid meat and meat products [19] and, as a corollary, could therefore impact defecation frequency.

Several authors emphasized that vegetarian populations differ from omnivorous populations with regard to bowel habits and defecation patterns. However, the number of studies investigating these associations is limited, and up-to-date trials in current vegetarian populations are lacking. The fact that vegetarian populations around the globe also differ significantly with regard to their actual dietary patterns also warrants consideration in this discussion. As such, it appears of paramount importance to investigate bowel health across different vegetarian populations.

We conducted a serial cross-sectional analysis of data from the National Health and Nutrition Examination Survey (NHANES) to characterize bowel health in U.S. vegetarians. The aims of this cross-sectional study were two-fold: (a) to investigate the prevalence of constipation in a well-characterized sample of U.S. vegetarians [20] and (b) to gain additional insights into defecating function in this particular cohort.

## 2. Materials and Methods

### 2.1. The NHANES

The NHANES is a cross-sectional, complex population-based survey with multiple components (including in-person interviews, laboratory and clinical examinations) [21]. NHANES is administered by the National Center for Health Statistics (NCHS) of the Centers for Disease Control on Prevention (CDC) [22]. The survey examines a nationally representative sample of approximately 5000 individuals annually. The sample for the survey is selected to represent the U.S. population of all ages. To produce reliable statistics and population estimates, NHANES oversamples certain groups, e.g., individuals aged 60 or older, African Americans and Hispanics.

One of the key NHANES features is the complex multistage, stratified, clustered and probability sampling design [23] that warrants special attention when analyzing NHANES data [24]. When analyzing NHANES data, weighting is of utmost importance to make reliable estimates that are representative of the U.S. civilian non-institutionalized population [25]. For this particular analysis, we appended 2 consecutive NHANES survey cycles (2007–2008 and 2009–2010). These cycle were chosen because both included a (self-reported) “vegetarian status” variable (see below) that divided the cohort in two groups: vegetarians and non-vegetarians. All NHANES survey protocols were approved by the NCHS Research Ethics Review Board.

### 2.2. Covariates

Data to describe both populations (vegetarians and non-vegetarians) stem from questionnaire-based personal interviews at participants’ homes, followed by visits at a NHANES mobile examination center (MEC) [26]. NHANES interviews cover demographic, dietary, socioeconomic and other health-related questions. For the purpose of this analysis, we merged demographic data, examination data and questionnaire data with dietary data.

Demographic data included gender (female/male), age (in years) and ethnicity [27,28]. Race/ethnicity comprised five categories including “Mexican American,” “non-Hispanic White,” “non-Hispanic Black,” “other Hispanic” and “Other Race” (includes mixed race). In addition, we investigated annual household income and marital status. The latter comprised three categories, including “married/living with a partner,” “widowed/divorced/separated” and “never married.” Vegetarian status was self-reported and based on the question: “do you consider yourself to be a vegetarian?”

Dietary data included daily calorie intake as well as macronutrient intake, fiber intake, caffeine intake, alcohol intake and daily aggregates of water (moisture) [29,30]. The latter included all moisture present in foods and beverages, including tap and bottled waters consumed as beverages. A previous analysis of NHANES vegetarians revealed that this cohort consumed, on average, significantly fewer calories than their omnivorous counterparts [20]. As such, we made use of a commonly used energy adjustment method to account for these difference in energy intake. We expressed nutrient density as intake (in gram or milligram) per 1000 kcal. Adjustment for total energy intake is frequently performed in epidemiological studies to minimize extraneous variation and to control for confounding factors [31,32]. The study by Juan and colleagues already included a detailed food group analysis in U.S. vegetarians [20], and as such we refrained from repeating this part. The aforementioned subset of dietary parameters was chosen because all variables (fiber, moisture, fat intake, etc.) have been previously associated with constipation and bowel health in general [33,34]. All dietary data were obtained from the dietary interview component [29,30]. This module uses a computerized 24 h dietary recall method to estimate energy and nutrient intake for all participants. A detailed description of the dietary data collection instrument (the so-called Automated Multiple Pass Method (AMPM)) can be obtained from the homepage of the U.S. Department of Agriculture [35]. For interested readers, the examination protocol and data collection methods are fully documented in the NHANES dietary interviewer’s procedure manuals [36].

Examination data were limited to anthropometric measurements including body mass index (BMI). We categorized BMI into for commonly used groups: obesity (BMI ≥ 30), overweight (BMI 25–29.99), normal weight (BMI 18.5–24.99) and underweight (BMI ≤ 18.49). All body measurement data were collected by trained health technicians with the support of a recorder during the body measurement examination [37,38].

### 2.3. Bowel Health Assessment

In order to explore the prevalence of constipation and to assess defecating function in U.S. vegetarians, we used data from the Bowel Health section of the mobile exam center (MEC) interview. The Bowel Health Questionnaire (variable name prefix BHQ) provides personal interview data on defecating function and fecal incontinence for adults age 20 years and older [39,40]. First administered in the NHANES in 2005, the 2007/2008 component comprised 6 questions. Based on the Fecal Incontinence Severity Index (FISI) by Rockwood et al. [41], 4 questions assessed adult incontinence leakage. The symptoms composing the index are incontinence of liquid stool, solid stool, gas and mucus [39]. The questions apply a type x frequency matrix to obtain the subject’s perception of symptom severity.

We assessed stool frequency using the following interview question: “How many times do you have a Bowel Movements (BM) per week?” We recorded BMs into five sub-groups following the approach by Mitsuhashi and colleagues [42], which comprised <3, ≥3–7, ≥8–14, ≥15–21 and ≥21 BMs per week. All categories were determined based on logical cutpoints (≥3–7 = up to one BM per day; ≥8–14 = one to two BMs per day; ≥15–21 = two to three BMs per day; ≥21 more than 3 BM per day).

An additional question included the Bristol Stool Form Scale (BSFS) [43]. The stool consistency scale has been used in a series of clinical studies to assess stool form in several gastrointestinal disorders [44]. BSFS demonstrated substantial validity and reliability [45]. Stool consistency was assessed using BSFS (BSFS: range from type 1–7). Participants were asked to define their stool by recording the number type that corresponded to their usual/most common stool type. BSFS demonstrated fair correlation with colonic transit time (CTT) and a BSFS of 1–2 has been suggested as a surrogate for delayed CTT in Westerners [46]. On the other hand, studies also found the BSFS score to correlate with the severity of incontinence [47]. Based on these findings, we defined constipation and diarrhea based on self-reported typical stool type. Following the approach of Wilson [48], stool type 1 (separate hard lumps, similar to nuts) and stool type 2 (sausage-like, but lumpy) on the BSFS were defined as constipation. Stool type 6 (fluffy pieces with ragged edges, a mushy stool) and stool type 7 (watery, no solid pieces) were defined as diarrhea.

Fecal incontinence (FI) was defined based on a previous study by Ditah and colleagues [49]. FI was defined as any involuntary loss or accidental leakage of mucus, liquid or solid stool during the past 30 days. The employed definition of FI did not include gas leakage, which we analyzed separately. Accidental bowel leakage was defined as leaking from the bowel or intestines that cannot be controlled.

### 2.4. Statistical Analysis

All statistical analyses were conducted using STATA version 14 (StataCorp., College Stadion, TX, USA). We used the “svyset” and “svy” commands for all statistical procedures to properly account for population weights and the complex NHANES survey design characteristics. These weights take into account unequal probabilities of selection resulting from the sample design, non-response and planned over-sampling of the elderly; non-Hispanic Blacks; and Mexican Americans [50]. Following NHANES guidelines, we generated a 4-year weight (2007–2010) to obtain weighted percentages adjusted to the US adult population. All variables were compared between self-identified vegetarians and non-vegetarians.

We described normally distributed variables with their mean and standard error in parentheses, whereas non-normally distributed variables are presented as median and interquartile range in parentheses. To describe categorical variables, we reported the number of observations (n) as well as weighted proportions (with their corresponding standard error) in parenthesis.

In order to make statistically valid population inferences from sample data, we computed standard errors using procedures that took into account the complex nature of the sample design [51]. Continuous (normally distributed) variables were compared by using appropriate sample weights for two-sample Student’s t-tests. For categorical variables, we used STATA’s design-adjusted Rao-Scott test (a design-adjusted version of the Pearson chi-square test) to explore potential associations between vegetarian status and the respective variables. Statistical significance was determined at α = 0.05. All tests for statistical significance were two-sided.

To determine the reliability of estimated proportions, we followed the strict recommendations of NCHS [52]. In order to assess the reliability of a proportion, we used the postestimation command “kg_nchs” in STATA [53]. Following the svy: proportion command “kg_nchs” allows users to calculate Korn–Graubard Confidence Intervals (CI) and to display a series of three dichotomous flags that show if NCHS standards are met. The command indicates whether the respective proportion (p) (and the complementary proportion (1-p)) meets all presentation standards and is considered reliable. Proportions that did not meet the NCHS standard (“unreliable proportions”, e.g., when the standard error exceeded 30% of the proportion estimate or when the relative CI width exceeded 130% of the proportion) were clearly marked for all tables.

## 3. Results

The present study included 9743 individuals (unweighted) with a complete dataset. Of these, n = 9531 were non-vegetarians and n = 212 were self-identified vegetarians. Table 1 shows demographic, anthropometric and other characteristics of our sample. The weighted vegetarian subpopulation comprised significantly more females compared to the non-vegetarian group (68.24% vs. 50.82%, *p* < 0.001). Although vegetarians tended to be slightly younger, we observed no significant differences with regard to mean age in both groups (*p* = 0.195).

Vegetarian status was not independent of ethnicity and education level (*p* < 0.001 and *p* = 0.001), as assessed by the design-adjusted Rao–Scott test (Table 1). The weighted proportion of vegetarians with a college degree (or higher) was significantly larger as compared to non-vegetarians (42.36% vs. 27.02%, *p* = 0.006). In contrast, the weighted proportion of individuals with “only” a high school degree was significantly higher in the non-vegetarian group (24.17%).

In addition to that, our results suggest no association between vegetarian status and marital status and annual household income, respectively (*p* = 0.230 and 0.462). The weighted proportion of current smokers, however, was significantly smaller in the vegetarian group (10.91% vs. 21.54%). Furthermore, vegetarian status was also significantly associated with body mass index (*p* = 0.001). The weighted proportion of individuals with a normal weight was significantly higher in the vegetarian group (43.59% vs. 28.57%, *p* = 0.001), whereas the proportion of obese individuals was significantly smaller (18.60% vs. 36.14%, *p* < 0.001).

When discussing these particular results, we believe it is important to highlight again that we followed the strict recommendations of NCHS to determine the reliability of the (weighted) estimated proportions [52]. In light of the limited number of observations with a full dataset in the vegetarian group (*n* = 212), some of the reported estimated proportions should be considered unreliable per NHCS analytic guidelines (e.g., the weighted proportion of underweight vegetarians, where the standard error exceeded 30% of the proportion estimate).

Table 2 compares nutrient intake by vegetarian status. Vegetarians consumed significantly fewer calories compared to non-vegetarians (1956.01 vs. 2185.96 kcal/d, *p* = 0.024). Energy-adjusted protein intake and fat intake were significantly higher in the non-vegetarian group, whereas energy-adjusted carbohydrate intake was higher in the vegetarian group. Of note, we also found a significant difference with regard to fiber intake (*p* < 0.001). Vegetarians consumed, on average, 21.33 g of fiber per day, whereas non-vegetarians consumed on average 16.43 g of fiber per day. We also observed a significant difference with regard to daily aggregates of water (including all moisture present in foods and beverages) between both groups. The latter was significantly higher in non-vegetarians (3042.78 g/d vs. 2811.15 g/d, *p* = 0.045).

Table 3 displays the examined bowel health items by vegetarian status. Our data suggest that all examined items (including BSS assessment, bowel movement frequency, fecal incontinence and bowel leakage) were independent of vegetarian status.

We observed no significant difference in the weighted proportion of vegetarians and non-vegetarians with a normal stool pattern (85.46% vs. 85.60%, as assessed by the BSFS). The number of self-identified vegetarians with a BSFS type 1 or type 2 stool was too small to predict a reliable weighted proportion (unweighted number of observations: *n* = 6 and *n* = 10, respectively). Importantly, almost 7.5% of the weighted non-vegetarian population suffered from constipation and approximately 7% suffered from diarrhea.

The weighted proportion of vegetarians with 3–7 bowel movements per week and 8–14 bowel movements per week was 61.46% and 28.73% (Table 3). Comparable numbers were observed in the non-vegetarian population. The weighted proportion of individuals with less than three bowel movements per week in the non-vegetarian group was almost 3.5%. Again, the data distribution did not allow for a reliable prediction of that particular proportion in the vegetarian group (unweighted number of observations: *n* = 4).

Ultimately, we observed no significant intergroup differences in the weighted proportion of individuals with bowel leakage (gas). The weighted proportion of vegetarians “never” leaking gas was slightly higher; however, the results were not statistically significant. The weighted proportion of individuals with fecal incontinence was also slightly higher in the vegetarian group; yet again, the results were not statistically significant (Table 3).

## 4. Discussion

The present study sought to investigate bowel health and constipation prevalence in a self-identified vegetarian population from NHANES (2007–2010). Previous studies revealed significant differences with regard to form and stool consistency as well as bowel movement frequency and constipation prevalence between vegetarians and non-vegetarians. Surprisingly, the present study could not confirm the aforementioned findings.

Our study revealed no associations between vegetarian status and all examined bowel health items, including stool consistency, bowel movement frequency and constipation prevalence. This comes at a surprise and warrants a further discussion as well as a closer look at the nutrient intake in both groups.

A plant-based diet contains substantially larger amounts of dietary fibers than non-vegetarian diets [11]. In 1986, Davies et al. investigated bowel function measurements of individuals with different eating patterns [15]. Fifty-one individuals (ten women and seven men who habitually consumed an omnivorous, vegetarian or vegan diet) were examined. Vegans had the highest fiber intake (47 g), followed by vegetarians (37 g) and those consuming an omnivorous diet (23 g). While mean gut transit time was comparable across all three groups, vegans passed more frequent and softer stools. Lacto-Ovo-Vegetarians passed stool more frequently (1.2 ± 0.5 stools per day) than non-vegetarians (1.0 ± 0.2 stools per day). The highest frequency was observed in vegans (1.7 ± 0.9 stools per day) The authors also reported that increasing dietary fiber was associated with a shorter transit, more frequent stools and softer forms. Compared to our cohort, the vegetarian cohort in the study by Davies et al. [15] consumed substantially more fiber (37 g/d vs. 21.33 g/d). This might (partially) explain why the authors observed significant intergroup differences, while we did not.

Another cross-sectional analysis from the United Kingdom using data from the European Prospective Investigation into Cancer and Nutrition, Oxford cohort (EPIC–Oxford), demonstrated comparable trends [54]. Sanjoaquin et al. analyzed bowel movement frequency in 20,630 men and women aged 22–97 years. Compared with participants who regularly ate meat (9.5 in men, 8.2 in women), mean bowel movement frequency was higher in vegetarians (10.5 in men, 9.1 in women) and especially in vegans (11.6 in men, 10.5 in women) [54]. Again, the authors observed significant positive associations between bowel movement frequency and intakes of dietary fiber. In this study, the odds of having daily bowel movements also increased with increasing fiber intake for both genders, with an odds ratio in the highest compared with the lowest intake category of 2.00 (95% CI: 1.38–2.90) in men and 1.43 (95% CI: 1.24–1.64) in women. Again, unlike in our study, significant intergroup differences between plant-based eaters and omnivores were found.

Significant differences between vegetarians and non-vegetarians were also found in an Indian study by Panigrahi and colleagues [55]. Vegetarians tended to pass stool more frequently than non-vegetarians (11.8 ± 4.5 vs. 11.3 ± 4.7; *p* < 0.05). Unfortunately, the exact amount of fiber intake per day was not assessed by the authors.

The authors of all three aforementioned studies found significant intergroup differences between vegetarians and non-vegetarians. Surprisingly, our study could not confirm this. The reasons for this warrant further discussion. At first, it is noteworthy that fiber intake in NHANES vegetarians is rather low (approximately 21 g/d) for this particular eating pattern. Self-identified NHANES vegetarians consumed a more plant-based diet than the U.S.: general population but differed from other “classical” (highly adherent) vegetarian populations in that they also occasionally consumed meat and fish [56].

As such, vegetarians in other cohorts, such as the French NutriNet-Santé cohort [57] or a comparable Belgian cohort, consumed substantially more fiber [58]. Vegetarians in our cohort, in turn, did not meet the Institute of Medicine’s recommendations for fiber intake (adequate intake: 14 g/1000 kcal per day), highlighting once more America’s substantial fiber gap [59]. This could be an important factor when discussing our results, as fiber obviously increases stool frequency in most individuals (particularly in those with constipation) [60].

Moreover, vegetarians in our cohort consumed less daily aggregates of water (moisture) than omnivores (Table 2). This included all moisture present in foods and beverages, including tap and bottled waters consumed as beverages. An adequate intake of fluid is necessary to enhance the effect of fiber in stool frequency regulation [61,62]. The fact that vegetarians in our sample consumed significantly less moisture than non-vegetarians may also explain why we observed no significant intergroup differences. Most other studies that investigated stool morphology and defecation patterns in vegetarian cohorts did not routinely report total moisture intake [15,18,55]. Only Sanjoaquin et al. reported fluid intake (including water and juices, but excluding milk, tea, coffee, soft drinks and alcoholic beverages) [54]. As such, a comparison to the other available studies remained difficult.

Third, Juan et al. already emphasized that a modest proportion of self-identified vegetarians in the NHANES occasionally reported consuming some type of animal products, such as meat, poultry and/or seafood [20]. As discussed before, some authors observed an inverse dose relationship between the amount of meat in diet and the frequency of defecation [18,63]. The fact that a modest proportion of self-identified vegetarians in this sample consumed meat may have also contributed to the non-significant intergroup differences with regard to defecation.

Finally, we acknowledge that some of our proportion estimates must be considered “unreliable” with regard to recent NCHS guidelines [52], and as such we were unable to reliably estimate the proportion of U.S. vegetarians suffering from constipation. Thus, in light of the low (unweighted) number of vegetarians with constipation (or BSFS stool type 1 or 2), we could not reliably answer all of our study questions.

Nevertheless, our study draws upon a number of strengths. First, we present a nationally representative and large dataset (National Health and Nutrition Examination Survey) in a field (defection and stool patterns in vegetarians) that received comparably little attention in the past years. Second, the employed methods and tools (including BSFS and FISI) demonstrated substantial validity and reliability in previous studies, which is important when comparing different populations. Third and probably most important, the present study reminds us once more of the fact that not all plant-based diets are created equal [64] and that vegetarian populations differ substantially around the globe [65]. This is particularly important with regard to fiber intake in our vegetarian cohort, which was well below the fiber intake in other non-US vegetarian cohorts [57,58,66]. As such, reporting our results appears important and shows that being a self-reported vegetarian does not equal adequate fiber intake per sé.

At the same time, our study has several weaknesses worth mentioning. Despite the large, population-based sample, we identified a relatively small number of individuals with a complete dataset that identified themselves as vegetarians (*n* = 212). Our sample may be extrapolated to represent 2.12% of the U.S. vegetarian population (weighted), which is slightly less than the estimated vegetarian population in 2008 (3.2%) in another study [67]. Unfortunately, the small sample size limits the precision of our estimates. In fact, estimates for some proportions in the vegetarian group should be considered unreliable per NCHS analytic guidelines (August 2017), as the standard error estimate exceeded 30% of the proportion estimate (or because the relative confidence interval width exceeded 130% of the proportion). This applies for several proportions (e.g., BSFS Type 1 and 2, BSS-based constipation and some proportions of the bowel movement frequency variable), and we clearly acknowledge this limitation of our study. Usage of STATA’s postestimation command “kg_nchs” enabled us to clearly identify the estimated proportions where this was the case. As such, not all our estimates are reliable and we had to focus our discussion on the ones that fulfilled the strict NCHS criteria.

Nevertheless, we refrained from reporting unweighted proportions (a frequently encountered alternative approach) and followed NCHS guidelines that mandate the use of sampling weights and sample design variables to obtain unbiased estimates and accurate standard errors. As such, some findings must be interpreted with caution. Yet, we believe it is important to report this data, highlighting the need for additional studies to characterize bowel health in U.S. vegetarians with regard to the few items where uncertainty remains.

We also acknowledge that our data are cross sectional in nature, and no causal inference can be drawn from this type of dataset. The fact that vegetarian status was self-reported may introduce a certain bias that we discussed elsewhere in great detail [68]. In addition to that, it is well known that several self-reported vegetarians in NHANES also consumed meat [20]. One may consider this additional limitation that could contribute to non-significant differences in defecation patterns between both groups. Moreover, it is of utmost importance to highlight that the employed dataset dates back to 2007–2010. It is a challenge to compare this sample to “modern” vegetarian populations because “nutritional awareness” has (potentially) increased over the past years among vegetarian cohorts [69]. It is not inconceivable that vegetarians currently pay more attention to adhere to a diverse plant-based diet, including sufficient amounts of fiber.

Ultimately, in light of the small available sample size, we did not stratify our results by gender. Given that some studies reported significant differences between females and males with regard to bowel habits [42], this would may have enhanced the quality of our data report. On the flipside, the sample size of the vegetarian groups already appears too low to reliably estimate proportions in even smaller groups.

## 5. Conclusions

To the best of our knowledge, we present the first study that particularly investigated bowel health in a cohort of NHANES vegetarians (2007–2010). The surprising lack of associations between vegetarian status and bowel health items could be explained by the relatively low fiber intake (and the lower moisture intake) in the vegetarian group. The fact that even the vegetarian cohort in this sample did not meet the Institute of Medicine’s recommendations for daily fiber intake demonstrates once again America’s distressing fiber gap. In comparison with other vegetarian cohorts, U.S. vegetarians consumed substantially less fiber. The relatively small sample size, however, warrants caution, and additional studies are required.

## Figures and Tables

**Table 1 nutrients-14-00681-t001:** Subgroup analysis of demographic, anthropometric and clinical characteristics by vegetarian status in individuals aged 20 years or older.

Demographic, Anthropometric and Clinical Characteristics	Non-Vegetarians	Vegetarians	*p* ^a^
	*n* = 9531	*n* = 212	
**Gender**			<0.001
Female	4785 (50.82 (0.44))	130 (68.24 (3.62)) ^a^	
Male	4746 (49.18 (0.44))	82 (31.76 (3.62)) ^a^	
**Age** (years)	46.93 (0.36)	44.56 (1.83)	0.195
Ethnicity			<0.001
Mexican American	1659 (8.25 (1.31))	34 (6.52 (1.68))	
Other Hispanic	956 (4.64 (0.81))	27 (4.93 (1.18)) ^c^	
Non-Hispanic White	4746 (71.01 (2.35))	91 (62.51 (6.81))	
Non-Hispanic Black	1810 (10.91 (1.08))	29 (7.73 (1.86))	
Other Race—Including Multi-Racial	360 (5.18 (0.51))	31 (18.30 (5.73)) ^a,c^	
**Education level**			0.001
	1067 (5.60 (0.43))	37 (8.38 (2.07))	
Less than 9th grade	1593 (12.92 (0.72))	22 (7.94 (1.95)) ^a^	
9–11th grade b	2303 (24.17 (0.86))	27 (12.77 (3.79)) ^a,c^	
High school graduate/GED or equivalent			
Some college or AA degree	2650 (30.28 (0.67))	57 (28.53 (3.88))	
College graduate or above	1918 (27.02 (1.28))	69 (42.36 (5.41)) ^a^	
**Marital Status**			0.23
Married/living with partner	5775 (64.46 (1.09))	120 (55.92 (5.25))	
Widowed/divorced/separated	2170 (18.25 (0.56))	55 (23.07 (4.08))	
Never married	1586 (17.30 (0.90))	37 (21.01 (4.62))	
**Annual household income**			0.462
Under $20,000	2071 (14.44 (0.84))	51 (17.06 (4.14))	
Over $20,000	7460 (85.56 (0.84))	161 (82.94 (4.14))	
**Smoking Status**			0.036
Never smoker	4991 (53.77 (1.17))	136 (63.77 (5.27))	
Former Smoker	2407 (24.69 (0.78))	55 (25.32 (3.75))	
Current Smoker	2133 (21.54 (0.79))	21 (10.91 (2.98) ^a^	
**Body mass index (BMI)**			0.001
Underweight (BMI ≤ 18.49)	137 (1.43 (0.18))	5 (4.47 (2.21)) ^c^	
Normal weight (BMI 18.5–24.99)	2464 (28.57 (0.78))	83 (43.59 (4.32)) ^a^	
Overweight (BMI 25–29)	3237 (33.85 (0.69))	70 (33.34 (3.84))	
Obese (BMI ≤ 30)	3693 (36.14 (0.71))	54 (18.60 (3.69)) ^a^	

Legend for Table 1: *n* = number of observations with weighted proportions in parenthesis. ^a^ = indicates statistically significant differences in the proportions. ^b^ = includes 12th grade with no diploma. ^c^ = estimate considered unreliable per NHCS analytic guidelines.

**Table 2 nutrients-14-00681-t002:** Nutrient intake analysis by vegetarian status in individuals aged 20 years or older. *n* = number of observations. We present data as mean + standard error proportions in parenthesis.

Dietary Intake	Non-Vegetarians *n* = 9531	Vegetarians *n* = 212	*p* ^a^
Energy value (kcal)/day	2185.96 (17.62)	1956.01 (94.10)	0.024
Protein (g/1000 kcal)	39.32 (0.22)	33.18 (0.92)	<0.001
Carbohydrate (g/1000 kcal)	121.72 (0.60)	141.11 (2.47)	<0.001
Fat (g/1000 kcal)	37.40 (0.17)	33.20 (1.04)	<0.001
Fiber (gm/1000 kcal)	7.99 (0.11)	11.43 (0.48)	<0.001
Fiber (gm)	16.43 (0.28)	21.33 (1.21)	<0.001
Alcohol (g/d)	11.78 (0.572)	8.12 (2.41)	0.146
Caffeine (mg/d)	188.33 (5.91)	166.77 (24.06)	0.356
Moisture (g/d)	3042.78 (25.31)	2811.15 (17.55)	0.045

Legend for Table 2: Normally distributed variables are shown with their mean and standard error in parentheses.

**Table 3 nutrients-14-00681-t003:** Bowel health by vegetarian status in individuals aged 20 years or older.

Bowel Health Characteristics	Non-Vegetarians *n* = 9531	Vegetarians *n* = 212	*p* ^a^
Bristol Stool Scale			0.330
Type 1	211 (1.97 (0.12))	6 (4.16 (2.40)) ^b^	
Type 2	521 (5.36 (0.34))	10 (2.78 (1.17)) ^a,b^	
Type 3	2293 (26.22 (0.46))	46 (27.53 (3.59))	
Type 4	4899 (51.28 (0.92))	106 (47.86 (4.67))	
Type 5	818 (8.09 (0.42))	23 (10.06 (2.80))	
Type 6	688 (6.31 (0.31))	19 (7.25 (1.77))	
Type 7	101 (0.75 (0.11))	2 (0.35 (0.25)) ^b^	
BSS-based stool pattern			0.919
Constipation	732 (7.33 (0.34))	16 (6.94 (2.55)) ^b^	
Normal	8010 (85.60 (0.47))	175 (85.46 (3.65))	
Diarrhea	789 (7.07 (0.30))	21 (7.60 (1.80))	
Bowel Movement Frequency			0.774
<3/week	339 (3.41 (0.32))	4 (2.28 (1.42)) ^b^	
3–7/week	5846 (64.02 (0.65))	125 (61.46 (3.78))	
8–14/week	2677 (26.36 (0.59))	67 (28.73 (2.61))	
≥15–21/week	535 (4.98 (0.27))	13 (6.18 (1.98)) ^b^	
≥21/week	134 (1.21 (0.14))	3 (1.35 (0.99)) ^b^	
Bowel leakage: gas			0.668
2 or more times a day	964 (9.99 (0.64))	20 (8.95 (2.07))	
Once a day	831 (9.11 (0.46))	20 (9.19 (2.52))	
2 or more times a week	681 (7.38 (0.34))	18 (7.04 (1.82))	
Once a week	499 (5.92 (0.24))	13 (6.52 (2.32)) ^b^	
1–3 times a month	1100 (11.99 (0.39))	25 (7.80 (2.48)) ^b^	
Never	5456 (55.59 (0.86))	116 (60.50 (4.06))	
Fecal Incontinence (FISI)			0.085
Yes	227 (1.89 (0.17)	9 (3.76 (1.40)) ^b^	
No	9304 (98.11 (0.17)	203 (96.24 (1.40))	

Legend for Table 3: *n* = number of observations with weighted proportions in parenthesis. ^a^ = indicates statistically significant differences in the proportions. ^b^ = estimate considered unreliable per NHCS analytic guidelines.

## Data Availability

Data are publicly available online (https://wwwn.cdc.gov/nchs/nhanes/Default.aspx; accessed on 22 January 2022). The datasets used and analyzed during the current study are available from the corresponding author upon reasonable request.

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
