# Peer review of "Bowel Health in U.S. Vegetarians: A 4-Year Data Report from the National Health and Nutrition Examination Survey (NHANES)"

_nutrients, 2022, doi:10.3390/nu14030681_

Round 1
Reviewer 1 Report
First, I would like to congratulate the authors for taking up such an interesting and current topic.
Introduction line - 33 to 69
- The introduction is clear and well-funded, so I don't think there's anything to add.
Materials and Methods - 70 to 195
- The methodology is well written and explained, but I did not understand how the dietary data were collected - was in a direct or indirect ways? With food records or food frequency questionnaires? I think is important to explain that.
- The description about Bowel health assessment is ok.
Results - 195 to 276
Table 2 - it is not correct to write calories but energy value, because the calorie is a unit of measurement as is the joule. Therefore, where in table 2 they refer to calories (kcal)/day, there must be energy value (kcal)/day.
The way the results are described is well done and well understood
Discussion 277 to 389
In the discussion you should no mention results, so when you mentioned table 3 in line 284, it should not be there.
In line 299 you try to explain the intergroup differences, but you did not discuss why in your group the quantity of fibre is too low. Do you have any data about it?
Even using the adequate statistical tests I think that the sample of vegetarians is very small compared to the sample of non-vegetarians. Do you have data about the percent of vegetarians in US population? If so it was interesting to include it.
The way in which information on intake data was collected may also have some bearing on the results found. It would be interesting to include this point in the discussion of the article.
You data ar of an extreme importance, so I need to congratulate you all.
Author Response
Dear Reviewer,
We would like to thank you very much for careful and thorough reading of this manuscript and for the thoughtful comments and constructive suggestions, which help to improve the quality of this article. We made all the requested revisions to our original manuscript based on all the comments we received from you. All changes have been clearly marked in yellow and green color. We appreciate your valuable input and time. Please find attached our revisions.
Sincerely,
The authors

Reviewer 2 Report
Thank you very much for the opportunity to read this manuscript. The Authors analyzed parameters regarding the frequency and “quality” of bowel movements between omnivores and vegetarians. The research is methodologically correct and clearly described. The manuscript is also well written in English. The results of the study are surprising. However, the Authors accurately described the study’s limitations and the factors that could influence this result. The greatest limitation of this study seems to be the relatively old age of the data (2007-2010). It is difficult to compare them directly to the modern population because “nutritional awareness” has increased significantly in these 15 years (probably even more attention is paid to the amount of fiber and water in the diet). This should be included in the discussion.
Below, the Authors will find some other minor comments on the manuscript.
Line 40: Perhaps it is worth briefly referring to the reasons for limiting dietary fiber.
Line 186 and 228: Unnecessarily repetition of the (NCHS) abbreviation explanation.
Line 328: Have other studies compared the amount of water consumed by vegetarians?
Author Response
Dear Reviewer,
We would like to thank you very much for careful and thorough reading of this manuscript and for the thoughtful comments and constructive suggestions, which help to improve the quality of this article. We made all the requested revisions to our original manuscript based on all the comments we received from you. All changes have been clearly marked in yellow and green color. We appreciate your valuable input and time. Please find our detailed response attached.
Sincerely
The authors
